# The Role of Biomarkers and Scores in Describing Urosepsis

**DOI:** 10.3390/medicina59030597

**Published:** 2023-03-17

**Authors:** Mădălin Guliciuc, Daniel Porav-Hodade, Bogdan-Calin Chibelean, Septimiu Toader Voidazan, Veronica Maria Ghirca, Adrian Cornel Maier, Monica Marinescu, Dorel Firescu

**Affiliations:** 1Clinical Emergency County Hospital “Sf. Ap. Andrei”, 800578 Galați, Romania; guliciuc.madalin@gmail.com; 2Faculty of Medicine and Pharmacy, Dunarea de Jos University, 800008 Galați, Romania; 3Faculty of Medicine and Pharmacy, “George Emil Palade” University of Medicine, Pharmacy, Science and Technology of Targu Mures, 540139 Târgu Mureș, Romania; 4Emergency Military Hospital Galati, 800150 Galați, Romania

**Keywords:** urinary tract infection, sepsis, diagnosis, biomarkers, sensitivity and specificity

## Abstract

*Background and Objectives*: Patients with urinary tract obstruction (UTO) and systemic inflammatory response syndrome (SIRS) are at risk of developing urosepsis, whose evolution involves increased morbidity, mortality and cost. The aim of this study is to evaluate the ability of already existing scores and biomarkers to diagnose, describe the clinical status, and predict the evolution of patients with complicated urinary tract infection (UTI) and their risk of progressing to urosepsis. *Materials and Methods*: We conducted a retrospective study including patients diagnosed with UTI hospitalized in the urology department of” Sfântul Apostol Andrei” County Emergency Clinical Hospital (GCH) in Galati, Romania, from September 2019 to May 2022. The inclusion criteria were: UTI proven by urine culture or diagnosed clinically complicated with UTO, fever or shaking chills, and purulent collections, such as psoas abscess, Fournier Syndrome, renal abscess, and paraurethral abscess, showing SIRS. The exclusion criteria were: patients age < 18 years, pregnancy, history of kidney transplantation, hemodialysis or peritoneal dialysis, and patients with missing data. We used the Sequential (Sepsis-Related) Organ Failure Assessment (SOFA) and qSOFA (quick SOFA) scores, and procalcitonin (PCT) to describe the clinical status of the patients. The Charlson Comorbidity Index (CCI) was used to assesses pre-existing morbidities. The hospitalization days and costs and the days of intensive care were considered. Depending on the diagnosis at admission, we divided the patients into three groups: SIRS, sepsis and septic shock. The fourth group was represented by patients who died during hospitalization. *Results*: A total of 174 patients with complicated UTIs were enrolled in this study. From this total, 46 were enrolled in the SIRS group, 88 in the urosepsis group, and 40 in the septic shock group. A total of 23 patients died during hospitalization and were enrolled in the deceased group. An upward trend of age along with worsening symptoms was highlighted with an average of 56.86 years in the case of SIRS, 60.37 years in the sepsis group, 69.03 years in the septic shock, and 71.04 years in the case of deceased patients (*p* < 0.04). A statistically significant association between PCT and complex scores (SOFA, CCI and qSOFA) with the evolution of urosepsis was highlighted. Increased hospitalization costs can be observed in the case of deceased patients and those with septic shock and statistically significantly lower in the case of those with SIRS. The predictability of discriminating urosepsis stages was assessed by using the area under the ROC curve (AUC) and very good specificity and sensitivity was identified in predicting the risk of death for PCT (69.57%, 77.33%), the SOFA (91.33%, 76.82%), qSOFA (91.30%, 74.17%) scores, and CCI (65.22%, 88.74%). The AUC value was best for qSOFA (90.3%). For the SIRS group, the PCT (specificity 91.30%, sensitivity 85.71%) and SOFA (specificity 84.78%, sensitivity 78.74%), qSOFA scores (specificity 84.78%, sensitivity 76, 34%) proved to be relevant in establishing the diagnosis. In the case of the septic shock group, the qSOFA (specificity 92.5%, sensitivity 82.71%) and SOFA (specificity 97.5%, sensitivity 77.44%) as well as PCT (specificity 80%, sensitivity 85.61%) are statistically significant disease-defining variables. An important deficit in the tools needed to classify patients into the sepsis group is obvious. All the variables have an increased specificity but a low sensitivity. This translates into a risk of a false negative diagnosis. *Conclusions*: Although SOFA and qSOFA scores adequately describe patients with septic shock and they are independent prognostic predictors of mortality, they fail to be accurate in diagnosing sepsis. These scores should not replace the conventional triage protocol. In our study, PCT proved to be a disease-defining marker and an independent prognostic predictor of mortality. Patients with important comorbidities, CCI greater than 10, should be treated more aggressively because of increased mortality.

## 1. Introduction

In daily urological practice, urinary tract infections (UTIs) complicated with obstructions and their progression to urosepsis have become an increasingly common pathology with an important impact on the morbidity and mortality of patients. They are also involving more and more human and material resources for their management. The patients with urinary tract obstruction (UTO) and systemic inflammatory response syndrome (SIRS) are at risk of developing urosepsis, whose evolution involves increased morbidity, mortality, and costs.

Urosepsis is defined as life-threatening organ dysfunction due to the host abnormal response to infections originating in the urinary tract or male genitalia [1]. Septic shock is defined as a subgroup of sepsis with cellular metabolism abnormalities and circulatory disorders which causes a substantial increase in the mortality rate. Thus, a broader approach was used to differentiate septic shock from cardiovascular dysfunction associated with sepsis and to recognize the importance of abnormalities in cellular metabolism [2]. The multiple organ dysfunction syndrome (MODS) is characterized by dysfunction of two or more organs, requiring the intervention of the clinician to maintain homeostasis [3].

Urosepsis represents 9–31% of the total cases of sepsis [4]. Urosepsis is a serious condition having an overall mortality that can range from 7.5% to 30% [5]. In addition, morbidity, as well as the increased costs of managing this disease, make it important to assess it promptly and rigorously for the purpose of providing rapid access to treatment [6]. An early diagnosis and an appropriate treatment can reduce the hospitalization costs, morbidity, and mortality [7].

The diagnosis of urosepsis is time-consuming and can only be confirmed with blood culture during bacteremia and urine culture results [8]. The process of diagnosing these infections can take from 24 to 72 h based on the time required to obtain the culture results. Additionally, there may be false positive results because of contamination and false negative due to antibiotic administration before admission. Therefore, rapid and efficient diagnostic methods for discriminating urosepsis from complicated UTI are necessary. Starting from these premises, we analyzed data already accessible to the clinician that could generate a prompt and specific assessment of the patients’ status and that could be used in establishing the diagnosis early so that the patients could benefit from adequate treatment in a timely manner.

For a more accurate assessment of the patient, we used procalcitonin (PCT) [9,10,11] and scores already confirmed by previous studies, such as quick SOFA (qSOFA) [12], the Sequential (Sepsis-Related) Organ Failure Assessment (SOFA) [13], and Charlson Comorbidity Index (CCI) [14]. This study aims to evaluate the ability of already existing scores and biomarkers to diagnose, describe the clinical status, and predict the evolution of patients with complicated UTI and their risk of progressing to urosepsis.

## 2. Materials and Methods

We conducted a retrospective study including patients diagnosed with complicated UTI hospitalized in the urology department of “Sfântul Apostol Andrei” County Emergency Clinical Hospital (GCH), an 800-bed general hospital in Galati, Romania, from September 2019 to May 2022. The GCH stands in Galati City which is home for 250,000 citizens. It serves the Galati County region, which has a population of 450,000 people.

This study was approved by the ethics committee of GCH, Galati, Romania. (Number: 24363/2021).

### 2.1. Patients Selection

The inclusion criteria were: UTIs proven by urine culture or diagnosed clinically, complicated with urinary tract obstruction, fever or shaking chills, and purulent collections such as psoas abscess, Fournier Syndrome, renal abscess, and paraurethral abscess, showing SIRS.

SIRS corresponds to the presence of two of the following criteria:Fever, temperature above 38 °C, or hypothermia below 36 °C.Tachycardia—over 90/min.Tachypnea—over 20 breaths/min or partial pressure of carbon dioxide in the arterial blood (PaCO_2_) < 32 mm Hg.Leukocytosis > 12,000/mm^3^ or leukopenia < 4000/mm^3^, or the presence of immature cells in the periphery < 10% [8].

The exclusion criteria were: patients age < 18 years, pregnancy, history of kidney transplantation, hemodialysis or peritoneal dialysis, and patients with missing data.

### 2.2. Collected Factors

A rigorous clinical examination was performed before admission. Clinical data were collected: heart rate, blood pressure, respiratory rate, PaO_2_, temperature and Glasgow Coma Scale.

Blood and urine samples were collected upon admission, following the International Safety Standards [15]. Complete blood count (CBC), total bilirubin, creatinine, and PCT levels were checked upon admission to the hospital.

PCT levels were measured using the automatic analyzer VIDAS BRAHMS PCT package insert, according to the manufacturer’s instructions. The lower limit of detection of the assay was 0.05 ng/mL, and the functional assay sensitivity was 0.09 ng/mL.

Demographics, clinical and laboratory findings, and diagnostics were recorded. All patients’ medical records were reviewed, and the relevant clinical and biological data were collected. All these data were systematized using qSOFA, SOFA scores, and CCI.

### 2.3. Vadiable Definition

PCT is a marker of systemic inflammation and thus may help to predict bacteremia [9,10,11].

For a simpler, faster, and less resource-consuming initial assessment of patients at risk of sepsis, the qSOFA score was used, which incorporates cognitive impairment (Glasgow Coma Scale < 15), systolic blood pressure of 100 mm Hg or less and respiratory rate of 22/min or higher [12].

To systematically and objectively describe the clinical condition of the patients on admission, we used the SOFA score, which evaluates the respiratory, nervous, and circulatory systems; the hepatic and renal functions; and coagulation [13]. The utility of the score had previously been validated on large cohorts of critically ill patients.

To describe the health status of patients before this acute event, the CCI was used, which assesses pre-existing morbidities [14].

To evaluate the impact on the health system, the hospitalization days and costs (quantified in RON, the Romanian currency, RON 1 representing approximately EUR 0.2) and the days of intensive care were considered.

### 2.4. Patient Grouping According to the Disease Stage

Depending on the diagnosis at admission, we divided the patients into three groups: SIRS, sepsis and septic shock. The fourth group was represented by patients who died during hospitalization.

Patients in the SIRS group were patients with complicated UTIs who met SIRS criteria but did not have sepsis on admission according to the new sepsis definition. The sepsis group included patients with sepsis not showing organ dysfunction. Septic shock patients exhibited hypotension, despite adequate fluid resuscitation or MODS. Patients who died during hospitalization were included in the deceased group.

### 2.5. Statistical Analysis

The continuous variables were expressed by descriptive statistics as mean ± standard deviation (SD) or median and interquartile range [IQR (Q1–Q3)], while the categorical variables were summarized by absolute and relative frequencies. All continuous variables were checked for normality using the Kolmogorov–Smirnoff test.

Descriptive statistics were analyzed through the prism of deceased status, SIRS, sepsis, and septic shock. The variables that have a Gaussian distribution (ex. age), were interpreted by means and SD, and the Students’ *t*-test was applied. Variables without a Gaussian distribution: leukocytes (WBC), PCT, SOFA, qSOFA, intensive care days, hospitalization days, and hospitalization costs were analyzed by median and interquartile range [IQR (Q1–Q3)], and the Mann–Whitney test was applied.

The correlation between quantitative variables was assessed using Spearmans’ rho correlation when appropriate.

A ROC curve analysis was performed in order to evaluate discriminant accuracy and to find the cut-off values for studied variables. The cutoff level for each variable depending on the analyzed group represents the level for which the best values for sensitivity (the ability to correctly identify the positive diagnosis) and specificity (the ability to correctly identify the negative diagnosis) are simultaneously identified. For all two-sided statistical tests, the significance was achieved if the estimated significance level *p*-value ≤ 0.05. Statistical analysis was performed using MedCalc Software, Version 12.5.0.0.

## 3. Results

A total of 174 patients with complicated UTI were enrolled in this study. The average age was 61.4 ± 15.9 years (mean ± SD), 116 (66.7%) patients were from the urban environment, and 107 (61.5%) patients were male.

From this total, 46 (24.4%) were enrolled in the SIRS group, 88 (50.6%) in the sepsis group, and 40 (22.9%) in the septic shock group. A total of 23 (13.2%) patients died during hospitalization, thus being enrolled in the deceased group.

Analyzing the above table, we highlighted an upward trend of age along with worsening symptoms, having an average of 56.86 ± 17.22 years for the SIRS group, 60.37 ± 15.54 years for the sepsis group, 69.03 ± 12.72 years for the septic shock group, and 71.04 ± 11.03 for the deceased group. Students’ *t*-test was applied, and a correlation between age and disease severity proved to be statistically significant (*p* < 0.05) except for the sepsis group (*p* = 0.40) (Table 1).

Given the fact that age is a variable in calculating CCI and that it is more likely to have more comorbidities with advancing age, Spearmans’ rho correlation was used to highlight if there was a statistically significant correlation between age and CCI. This proved to be statistically significant (*p* < 0.001).

WBC count [median (IQR)] was analyzed using the Mann–Whitney test and proved to be statistically significant only in the case of the septic shock group [18,920.0 (12,395.0–28,140.0) × 10^6^/L] *p* < 0.01 and was not statistically relevant in the case of the SIRS [16,065.0 (12,050.0–19,310.0) × 10^6^/L] *p* = 0.79, the sepsis [16,360.0 (11,367.5–19,937.5) × 10^6^/L] *p* = 0.16, and the deceased [18,270.0 (12,582.5–24,550.0) × 10^6^/L] *p* = 0.24 groups (Table 1).

PCT median (IQR) increased according to the disease severity: 2.45 (0.70–3.20) ng/mL for the SIRS group, 9.6 (6.3–12.2) ng/mL for the sepsis group, 24.7 (13.5–32.0) ng/mL for the septic shock group, and 32.0 (10.38–32.0) for the deceased group. This proved to be statistically significant for all groups (*p* < 0.01) (Table 1).

SOFA score median (IQR) tended to rise with disease severity as follows: 3.0 (2.0–4.0) for the SIRS group, 6.0 (4.0–7.0) for the sepsis group, 9.5 (8.0–12.0) for the septic shock group, and 10.0 (8.25–12.0) for the deceased group. Even though this upward trend is evident, the Mann–Whitney test could not show it to be statistically significant in the case of the sepsis group (*p* = 0.78). For the other groups, the *p* value was less than 0.001, therefore statistically significant (Table 1).

The same pattern can be seen with the qSOFA score, median (IQR) having a higher value depending on the disease severity: 0.0 (0.0–0.0) for SIRS, 1.0 (0.0–1.75) for sepsis, 2.0 (2.0–3.0) for septic shock, and 3.0 (2.0–3.0) for deceased, but without being statistically significant for the sepsis group (*p* = 0.32). For the other groups, the *p* value was less than 0.001, hence statistically significant (Table 1).

CCI median (IQR) was shown to rise directly with sepsis stages, being: 4.0 (1.0–8.0) for the SIRS group, 7.0 (3.0–9.0) for the sepsis group, 9.0 (7.0–12.0) for the shock septic group, and 11.0 (9.2–13.0) for the deceased group. This proved to be statistically significant for all groups (*p* < 0.001) except the sepsis group (*p* = 0.46) (Table 1).

The number of intensive care days required for treating these patients was directly proportional to case severity. This was found to be statistically significant for all groups (*p* < 0.001) (Table 1).

Analyzing the number of hospitalization days [median (IQR)], an upward trend depending on the disease severity can be observed: 5.0 (3.0–8.0) for the SIRS, 9.0 (6.0–13.7) for the sepsis, and 15.0(8.0–22.5) for the septic shock. This did not apply to the deceased group [10.0 (6.25–18.7)] *p* = 0.22 (Table 1).

Increased hospitalization costs [median (IQR)] were shown in the case of deceased patients [RON 10,855.0 (6752.5–24,053.2), *p* < 0.001] and those with septic shock [RON 14,704.5 (7357.0–26,103.5), *p* < 0.001] and statistically significantly lower in case of those with SIRS [RON 2863.0 (1247.0–6833.0), *p* < 0.001] (Table 1).

Given the fact that the variables that describe the patient from an economic point of view are theoretically interdependent, we used Spearman’s rho to highlight whether this also applies for the patients in our study. Comparing intensive care day and hospitalization days with hospitalization costs, it returned *p* < 0.001 in both cases, thus confirming this hypothesis.

Spearmans’ rho correlation was used to assess correlations between quantitative variables. These data have been attached to Appendix A.

The predictability of discriminating complicated UTIs stages was assessed by using the area under the ROC curve (AUC).

All variables were compared with each other, one by one, for each individual group. These data were systematized in laborious tables that cannot be incorporated into the main text. It had been added to Appendix A.

Comparing the ROC curves for the variables of interest in the deceased group, a very good specificity and sensitivity was identified in predicting the risk of death for PCT (69.57%, 77.33%), SOFA (91.33%, 76.82%), qSOFA (91.30%, 74.17%) scores, and CCI (65.22%, 88.74%). The best AUC value were 90.3% for qSOFA (Figure 1, Table 2).

For the SIRS group, PCT (specificity 91.30%, sensitivity 85.71%), SOFA (specificity 84.78%, sensitivity 78.74%), and qSOFA scores (specificity 84.78%, sensitivity 76, 34%) proved to be relevant in establishing the diagnosis. The most relevant parameter was PCT with a cutoff value of 4.8 (AUC 93%) (Figure 2, Table 2).

We can observe that for patients in the sepsis group no variable, except PCT, can provide significant statistical values to attest to the diagnosis. All variables were found to have increased specificity at the expense of sensitivity. The best AUC value (59.3%) was shown for PCT with a specificity of 83.72% and sensitivity of 53.49% (Figure 3, Table 2).

In the case of the septic shock group, as expected, qSOFA (specificity 92.5%, sensitivity 82.71%), SOFA (specificity 97.5%, sensitivity 77.44%), and PCT (specificity 80%, sensitivity 85.61%) were statistically significant disease-defining variables (Figure 4, Table 2), the best AUC value being for the SOFA score (93.9%).

## 4. Discussion

The clinical aspect of UTIs can vary from a simple cystitis to septic shock and exitus. In daily practice, it is very important to assess the risk of disease progression and to promptly intervene when the situation requires it [2]. For this purpose, there are many tools to evaluate the patient. The already validated scores, such as SOFA and qSOFA, were used to evaluate the patient diagnosed with complicated UTIs [12,13], CCI to assess the patients’ performance status before this acute event [14], and PCT as a reliable indicator of bacterial infection and systemic impact [9,10,11].

WBC have been shown to be of low importance in evaluating patients with urosepsis and their mortality risk. Even if WBC can raise an alarm signal regarding the patients’ clinical status, being a statistically significant marker for patients in the septic shock group, they are not precise in the differential diagnosis between SIRS and sepsis. Moreover, comparing WBC with the other variables proved to be the least accurate in defining the disease. This is an important statement given that clinicians tend to place too much importance in day-to-day practice on the WBC count when evaluating a septic patient.

Analyzing the patients’ age, an increased risk of developing septic shock, and death can be seen at advanced age. Given the fact that age is a variable in the CCI calculation and that theoretically the elderly patient is prone to develop comorbidities, we evaluated the correlation between age and CCI. Using Spearman’s rank correlation coefficient, we demonstrated a statistically significant correlation between age and CCI (*p* < 0.001). CCI was not developed to evaluate patients in an acute event, this being statistically confirmed when we wanted to use it to differentiate the urosepsis stages. However, it is obvious that a high CCI value (cutoff value 10, AUC 86.3%) is associated with an increased mortality rate. Thus, it proves to be an independent risk factor for mortality. Yang et. al. started to assess the disease burden of sepsis and to test the usefulness of CCI and age as risk-adjusted hospital mortality predictors in patients with sepsis. They concluded that comorbidities and advanced age were some of the most important contributors to hospital mortality and resource utilization [16]. A multicenter, retrospective cohort study assessed patients hospitalized with sepsis in the ICU in 7 general hospitals in Israel over a period of 7 years and proved that mortality is correlated with underlying patients’ characteristics, including age and comorbidities [17].

Organ dysfunction is associated with high morbidity and mortality rates [18] and, as such, accounts for a high proportion of the ward budget [19]. Organ failure scores, such as the SOFA score, can help assess organ dysfunction or failure over time and are useful to evaluate morbidity [20]. Although this scoring system was developed to describe and quantify organ function and not to predict outcome, the obvious relationship between organ dysfunction and mortality has been demonstrated in several studies [21,22,23]. The SOFA score is an increasingly important tool in defining both the clinical condition of the individual patient and the response to therapies in the context of clinical trials [24]. The SOFA score was validated on large cohorts to be an independent mortality predictor [25]. This was also statistically significant in our study; it was shown to be correct for the positive diagnosis of SIRS and septic shock and an independent predictor of mortality (cutoff value 7). However, SOFA fails to classify patients into the sepsis group, having a high specificity of 91.95% but a low sensitivity of 32.58%.

For the qSOFA score, having fewer variables and being based on the patients’ clinical assessment rather than more precise paraclinical data, one would expect it to have lower sensitivity and specificity than the SOFA score. However, it turns out to be similar to the SOFA score, even superior as a predictor of mortality. In order to classify patients in the sepsis group, the qSOFA proved to have the same shortcomings as SOFA, having low sensitivity (20.93%) while maintaining high specificity (98.85%).

PCT, being a simple, cheap, and accessible test, proved to be very important, having the ability to correctly classify patients in disease groups. A PCT value less than or equal to 4.8 has the best ability to rule out urosepsis compared to the other variables (specificity 91.30%, sensitivity 85.71%). Although it does not have a very good sensitivity (53.49%), it proves to be the most accurate variable for classifying patients in the sepsis group with a specificity of 83.72% (AUC 59.3%). PCT accurately predicts the presence of bacteremia and bacterial load in patients with complicated UTI [26]. Some studies have shown that a PCT > 2 ng/mL has >90% specificity for sepsis or the progression to sepsis [27]. In our study, the cutoff value for PCT in discriminating sepsis was 4.8. The PCT levels were higher in our study for the following possible reasons: most of the patients with UTI were infected with Gram-negative organisms, which cause higher peak PCT values than infections caused by Gram-positive organisms [28]. Different observation times may have resulted in different PCT optimal cutoff values to diagnose sepsis [29], and different test kits and methods may have been used [30].

The septic shock group was relatively easy to diagnose. This is also valid for patients who died, SOFA, qSOFA, PCT and CCI being independent prognostic predictors of mortality.

Analyzing all the variables, an important deficit in the tools needed to classify patients into the sepsis group is obvious. All the variables have an increased specificity, but a low sensitivity. This translates into a risk of false negative diagnosis leading to the lack of adequate treatment. Askim et al. concluded in their study that qSOFA failed to identify two-thirds of the patients admitted to the emergency department with sepsis, and it should not replace the traditional triage system [31]. A study set out to assess the value of several disease severity scores for the prognostic assessment of sepsis; it concluded that SOFA and qSOFA scores cannot take the place of traditional evaluation in patients at risk of developing sepsis [32]. Given the fact that urosepsis is a pathology with possible severe repercussions, death included, and that there is a risk of misclassifying patients with severe UTIs, we must act promptly. An adequate initial (e.g., in the first hour) antibiotic therapy ensures improved outcome in septic shock [33,34] and is also critical in severe UTIs [35] as has been shown with other infection sites as well [36,37]. Empirical antibiotic therapy considers the expected bacterial spectrum, the institutional specific resistance rates, and the individual patients’ requirements [38,39,40]. If a complicating factor in the urinary tract warranting treatment is identified, control and/or removal of the complicating factor should follow in the first 6 h [41].

From a statistical point of view, urosepsis represents a burden for the health system, directly proportional to the severity of the case. The septic shock cases are associated with increased hospitalization time and costs, also requiring additional days of ICU. Sepsis is the greatest financial burden for hospitals and the leading cause of death in noncoronary ICU cases, contributing to 30–50% of all in-hospital deaths [42]. Urosepsis is a severe urological condition with a significant mortality rate. The patient with complicated UTI should benefit from a complex clinical and paraclinical examination in a timely manner so that the correct diagnosis can be established and (s)he can receive appropriate treatment [43]. Given the fact that urosepsis is a pathology that can be getting worse, early diagnosis and treatment are imperative and can reduce hospitalization costs, as well as morbidity and mortality. The population should be educated about the implications of this pathology and encouraged to seek specialized help at the first symptoms of UTI.

This study also had several limitations. It was a retrospective study, and therefore, bias was likely. Moreover, the scores used were calculated at admission, not having clinical data necessary to calculate other important scores for our study, such as National Early Warning Score (NEWS), Modified Early Warning Score (MEWS), etc. The data were collected from a single center. As such, it could differ from those of other centers. Additional prospective studies with larger populations involving multiple centers are necessary to validate our conclusions.

## 5. Conclusions

Although SOFA and qSOFA scores adequately describe patients with septic shock and they are independent prognostic predictors of mortality, they fail to be accurate in diagnosing sepsis. These scores should not replace conventional triage protocols.

In our study, PCT proved to be a disease-defining marker and an independent prognostic predictor of mortality.

Patients with important comorbidities, CCI greater than 10, should be treated more aggressively because of increased mortality.

## Figures and Tables

**Figure 1 medicina-59-00597-f001:**
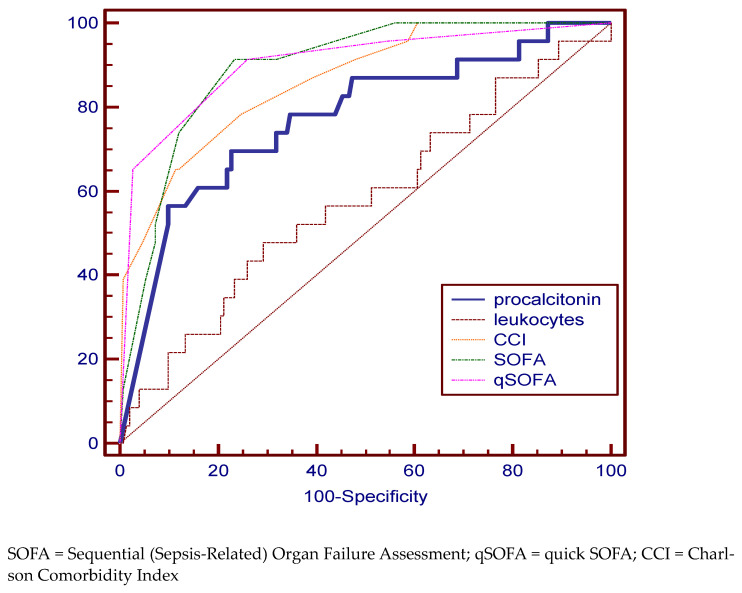
Pairwise comparison of ROC curves for the deceased group.

**Figure 2 medicina-59-00597-f002:**
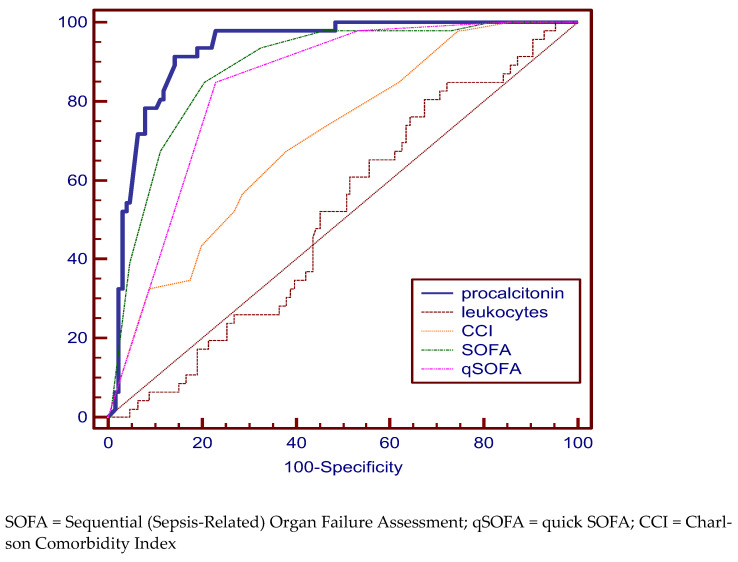
Pairwise comparison of ROC curves for the SIRS group.

**Figure 3 medicina-59-00597-f003:**
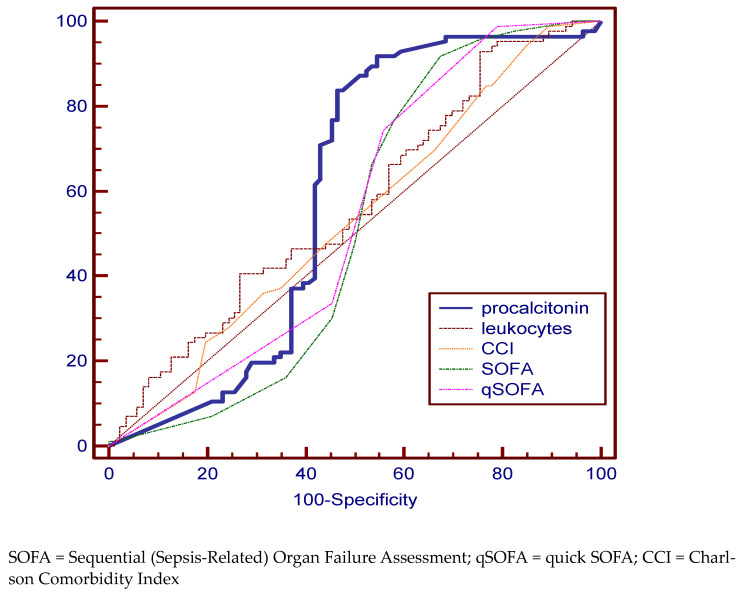
Pairwise comparison of ROC curves for the sepsis group.

**Figure 4 medicina-59-00597-f004:**
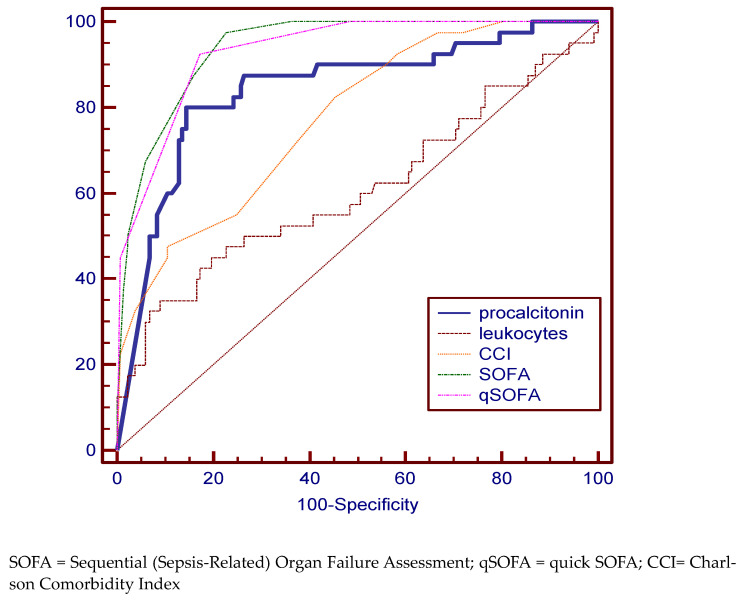
Pairwise comparison of ROC curves for the septic shock group.

**Table 1 medicina-59-00597-t001:** Bivariate analysis of variables according to studied groups.

	Deceased		SIRS		Sepsis		Septic Shock	
	Yes	No	Yes	No	Yes	No	Yes	No
No	23	151	46	128	88	86	40	134
Age (years) Mean ± SD	71.04 ± 11.03	59.91 ± 16.04	56.84 ± 17.22	62.97 ± 15.17	60.37 ± 15.54	62.40 ± 16.29	69.03 ± 12.72	59.10 ± 16.08
*p* value **	<0.001		<0.01		0.40		<0.0001	
leukocyte (10^6^/L) median (IQR)	18,270.0 (12,582.5–24,550.0)	16,360.0 (11,615.0–20,625.0)	16,065.0 (12,050.0–19,310.0)	16,470.0 (11,635.0–21,705.0)	16,360.0 (11,367.5–19,937.5)	16,530.0 (12,082.5–22,337.5)	18,920.0 (12,395.0–28,140.0)	16,315.0 (11,600.0–19,660.0)
*p* value *	0.24		0.79		0.16		<0.01	
Procalcitonin (ng/mL)	32.0 (10.38–32.0)	7.55 (2.70–12.2)	2.45 (0.70–3.20)	11.20 (7.25–22.32)	9.6 (6.3–12.2)	4.7 (1.7–22.1)	24.7 (13.5–32.0)	6.3 (2.70–10.6)
*p* value *	<0.0001		<0.0001		<0.01		<0.0001	
SOFA	10.0 (8.25–12.0)	5.0 (3.0–7.0)	3.0 (2.0–4.0)	7.0 (5.0–9.0)	6.0 (4.0–7.0)	5.0 (3.0–9.0)	9.5 (8.0–12.0)	4.5 (3.0–6.0)
*p* value *	<0.0001		<0.0001		0.78		<0.0001	
qSOFA	3.0 (2.0–3.0)	1.0 (0.0–2.0)	0.0 (0.0–0.0)	1.0 (1.0–2.0)	1.0 (0.0–1.75)	1.0 (0.0–2.0)	2.0 (2.0–3.0)	0.0 (0.0–1.0)
*p* value *	<0.0001		<0.0001		0.32		<0.0001	
CCI	11.0 (9.2–13.0)	6.0 (2.0–8.0)	4.0 (1.0–8.0)	8.0 (4.0–9.5)	7.0 (3.0–9.0)	7.0 (3/25–9/0)	9.0 (7.0–12.0)	6.0 (2.0–8.0)
*p* value *	<0.0001		<0.0001		0.46		<0.0001	
Intensive care days	5.0 (2.0–8.75)	0.0 (0.0–1.0)	0.0 (0.0–0.0)	1.0 (0.0–3.0)	0.0 (0.0–1.0)	1.0 (0.0–3.0)	3.5 (2.0–7.0)	0.0 (0.0–1.0)
*p* value *	<0.0001		<0.0001		<0.001		<0.0001	
Hospitalization days	10.0 (6.25–18.7)	8.0 (5.0–13.0)	5.0 (3.0–8.0)	10.0 (6.5–17.0)	9.0 (6.0–13.7)	7.0 (4.0–13.7)	15.0 (8.0–22.5)	8.0 (5.0–11.0)
*p* value *	0.22		<0.0001		0.12		<0.0001	
Hospitalization costs (Lei)	10855.0 (6752.5–24,053.2)	5329.0 (2230.2–9690.5)	2863.0 (1247.0–6833.0)	7309.0 (3416.5–14,994.0)	5836.0 (2351.7–9398.2)	6833.0 (2663.7–11,805.0)	14704.5 (7357.0–26,103.5)	4437.5 (1914.0–7967.0)
*p* value *	<0.001		<0.0001		0.40		<0.0001	

* Mann–Whitney test; ** Students’ *t*-test; SIRS = systemic inflammatory response syndrome SOFA = Sequential (Sepsis-Related) Organ Failure Assessment; qSOFA = quick SOFA; CCI = Charlson Comorbidity Index.

**Table 2 medicina-59-00597-t002:** Pairwise comparison of ROC curve.

Deceased					
	Cutoff value	Specificity % (95% CI)	Sensitivity % (95% CI)	AUC %	95% CI
procalcitonin	12.5	69.57 (47.1–86.8)	77.33 (69.8–83.8)	77.2	70.2 to 83.2
leukocytes	19,340 × 10^6^/L	47.83 (26.8–69.4)	70.86 (62.9–78.0)	57.6	49.9 to 65.1
CCI	10	65.22 (42.7–83.6)	88.74 (82.6–93.3)	86.3	80.2 to 91.0
SOFA	7	91.33 (72.0–98.9)	76.82 (69.3–83.3)	89.3	83.7 to 93.5
qSOFA	1	91.30 (72.0–98.9)	74.17 (66.4–80.9)	90.3	84.9 to 94.3
SIRS group					
procalcitonin	4.8	91.30 (79.2–97.6)	85.71 (78.4–91.3)	93.0	88.1 to 96.3
leukocytes	19,750 × 10^6^ L	80.43 (66.1–90.6)	32.28 (24.3–41.2)	51.1	43.4 to 58.8
CCI	6	67.39 (52.0–80.5)	61.42 (52.4–69.9)	70.3	62.9 to 77.0
SOFA	4	84.78 (71.1–93.7)	78.74 (70.6–85.5)	88.3	82.5 to 92.7
qSOFA	0	84.78 (71.1–93.7)	76.38 (68.0–83.5)	83.8	77.4 to 88.9
Sepsis Group					
procalcitonin	4.8	83.72 (74.2–90.8)	53.49 (42.4–64.3)	59.3	51.6 to 66.8
leucocytes	24,070 × 10^6^/L	93.10 (85.6–97.4)	24.43 (15.8–34.9)	56.8	49.0 to 64.3
CCI	11	94.25	15.12	53.6	45.9 to 61.2
SOFA	8	91.95 (84.1–96.7)	32.58 (22.8–43.5)	51.4	43.7 to 59.1
qSOFA	2	98.85 (93.8–100.0)	20.93 (12.9–31.0)	54.3	46.5 to 61.9
Septic Shock group					
procalcitonin	12.4	80.00 (64.4–90.9)	85.61 (78.4–91.1)	84.1	77.8 to 89.2
leukocytes	24,260 × 10^6^ L	35.00 (20.6–51.7)	90.98 (84.8–95.3)	60.7	53.0 to 68.1
CCI	6	82.5 (67.2–92.7)	54.89 (46.0–63.5)	77.4	70.4 to 83.4
SOFA	6	97.50 (86.8–99.9)	77.44 (69.4–84.2)	93.9	89.3 to 97.0
qSOFA	1	92.50 (79.6–98.4)	82.71 (75.2–88.7)	93.0	88.2 to 96.4

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
