# Peer review of "The Role of Biomarkers and Scores in Describing Urosepsis"

_medicina, 2023, doi:10.3390/medicina59030597_

Round 1
Reviewer 1 Report
General:
- Please carefully prepare your revised manuscript according to the authors' guidelines.
- The manuscript needs extensive English grammar and spelling revision is required before resubmission.
Abstract:
- In the Abstract and the article itself, define each abbreviation when first used—e.g. h urinary tract infections (UTIs)—and thereafter use the abbreviation alone without further explanation.
- Background: This section should be the shortest part of the abstract and should very briefly outline the following information: What is already known about the subject, related to the paper in question, and What is not known about the subject and hence what the study intended to examine.
- Methods: This section should contain enough information to enable the reader to understand what was done, and how. Please revise this section.
- Your study is retrospective. Please revise it.
- Result: I did not see any findings of your study. Please add this section.
- Please mention the conclusion section.
Keywords:
- The keywords should be Medical Subject Headings (MeSH®) Terms (http://www.ncbi.nlm.nih.gov/mesh).
Introduction
- Summarize the relevant literature on your topic.
- The study aim should be clearly explained in the last sentence of the introduction section.
Materials and Methods:
- "Uroculture" should be urine culture.
- The study design is not clear and should be revised to have sufficient details.
- "blood cultures" how the blood culture could confirm the diagnosis of UTI?
- sample size calculations should be mentioned.
- Study variables:
- In this section, you should mention all collected factors.
- You may add a new section with the title "variable definition" and you can define any factor involved in your study.
- The study outcome should be mentioned.
- Statistical analysis: Describe statistical methods in detail, with emphasis on the statistical strategy used to analyze the data.
- "The variables that have a Gaussian distribution" revise it to quantitive variables, etc…
- The tests used in comparison should be mentioned.
Results:
- This section should be started by Giving characteristics of study participants (eg demographic, clinical, social). Please revise this section.
- All tables' names should be revised.
- Table 1:
- The WBC count is a part of SIRS. What is your aim in mentioning it in table 1?
- Mean and SD should be in one column.
- In the (mean 16911.41) and (SD 7792.449), the SC variation is rich to the half value of the mean number, it may due to high variation or error in data analysis. Please review all results.
- Avoid mentioning "we" in this section.
- Please mention the result of all mentioned factors in your method.
- This table may revise to include and analyze all factors of your study (You may mention all factors in rows and the grouping in columns).
- Table 2: It is hard to follow, please revise it.
- The presentation of the obtained result is poorly written and needs to be rewritten with more careful preparation.
Discussion:
- This section is poorly written. For studies, your Discussion section should first reiterate briefly the results, then move to a discussion of your main findings, and finally, move to wider topics and comparison of your study with other research.
- The limitation of your study should be mentioned.
Conclusions:
- This section should summarize your main arguments and important findings. Please revised this section.
References
- Please correct your references according to the authors' guidelines.
- Journal names should be abbreviated to the Journals Database section in PubMed (http://www.ncbi.nlm.nih.gov/nlmcatalog/journals).
Author Response
Comments and Suggestions for Authors
General:
- Please carefully prepare your revised manuscript according to the authors' guidelines.
- The manuscript needs extensive English grammar and spelling revision is required before resubmission.
Abstract:
Thank you for giving enough resources to be so explicit, these arguments helped us to reformulate the Abstract section.
- In the Abstract and the article itself, define each abbreviation when first used—e.g. h urinary tract infections (UTIs)—and thereafter use the abbreviation alone without further explanation.
We have addressed to this issue.
- Background: This section should be the shortest part of the abstract and should very briefly outline the following information: What is already known about the subject, related to the paper in question, and What is not known about the subject and hence what the study intended to examine.
The patients with urinary tract obstruction (UTO) and Systemic Inflammatory Response Syndrome (SIRS) are at risk of developing urosepsis, whose evolution involve increased morbidity, mortality and costs. The aim of this study is to evaluate the ability of already existing scores to diagnose, describe the clinical status and predict the evolution of patients with complicated urinary tract infection (UTI) and their risk of progressing to urosepsis.
- Methods: This section should contain enough information to enable the reader to understand what was done, and how. Please revise this section.
- Your study is retrospective. Please revise it.
We conducted a retrospective study including patients diagnosed with UTIs hospitalized in the Urology Department of ” Sfântul Apostol Andrei ” County Emergency Clinical Hospital (GCH) in Galati, Romania, from September 2019 to May 2022. The inclusion criteria were: UTIs proven by urine culture or diagnosed clinically complicated with urinary tract obstruction, fever or shaking chills, and purulent collections such as psoas abscess, Fournier Syndrome, renal abscess, paraurethral abscess, showing SIRS. The exclusion criteria were: patients aged < 18 years, pregnancy, history of kidney transplantation, hemodialysis or peritoneal dialysis and patients with missing data. We used the Sequential (Sepsis-Related) Organ Failure Assessment (SOFA) score. For the a simpler, faster and less resource consuming initial assessment of patients at risk of sepsis, the qSOFA score (quick SOFA) was used. The Charlson Comorbidity Index (CCI) was used, which assesses pre-existing morbidities. The hospitalization days and costs and the days of intensive care were considered. Statistical analysis was performed using the MedCalc Software.
- Result: I did not see any findings of your study. Please add this section.
A total of 174 patients with complicated UTIs were enrolled in this study. From this total, 46 were enroled in the SIRS group, 88 in the Urosepsis group and 40 in the Septic Shock group. 23 patients died during hospitalization and were enrolled in the Deceased group.
An upward trend of age along with worsening symptoms was highlighted with an average of 56.86 years in the case of SIRS, 60.37 years in the Sepsis group, 69.03 years in the Septic Shock and 71.04 years in the case of deceased patients (p<0.04). Leukocytes count proved to be statistically significant in the case of sepsis (15986.52 x106/L) and septic shock (20555.40 x106/L) and were not statistically relevant in the case of SIRS (16872.22 x106/L) and deceased (19374.35 x106/L). A statistically significant association between procalcitonin and complex scores (SOFA, CCI and qSOFA) with the evolution of urosepsis was highlighted. Increased hospitalization costs can be observed in the case of deceased patients and those with septic shock and statistically significantly lower in the case of those with SIRS. The predictability of discriminating urosepsis stages was assessed by using the area under the ROC curve (AUC) and very good specificity and sensitivity can be identified in predicting the risk of death for procalcitonin (69.57%, 77.33%), the SOFA (91.33%, 76.82%), qSOFA (91.30%, 74.17%) scores and CCI (65.22%, 88.74%). The AUC value being best for qSOFA (90.3%). For the SIRS group the procalcitonin (specificity 91.30%, sensitivity 85.71%) and SOFA (specificity 84.78%, sensitivity 78.74%), qSOFA scores (specificity 84.78%, sensitivity 76, 34%) proved to be relevant in establishing the diagnosis. In the case of the septic shock group the qSOFA (specificity 92.5%, sensitivity 82.71%) and SOFA (specificity 97.5%, sensitivity 77.44%) as well as procalcitonin (specificity 80%, sensitivity 85.61%) are statistically significant disease-defining variables. An important deficit in the tools needed to classify patients into the Sepsis group is obvious. All the variables have an increased specificity, but a low sensitivity, this translates into a risk of false negative diagnosis.
- Please mention the conclusion section.
Although SOFA and qSOFA scores adequately describe patients with septic shock and they are independent prognostic predictors of mortality, they fail to be accurate in diagnosing sepsis. These scores should not replace conventional triage protocol. In our study PCT has proven to be a disease-defining marker and an independent prognostic predictor of mortality. Patients with important comorbidities, CCI greater than 10, should be treated more aggressively because of increased mortality
Keywords:
- The keywords should be Medical Subject Headings (MeSH®) Terms (http://www.ncbi.nlm.nih.gov/mesh).
Using MeSH database, we have chosen the following keywords: urinary tract infection, sepsis, diagnosis, biomarkers, sensitivity and specificity.
Introduction
- Summarize the relevant literature on your topic.
For this purpose, we added to this section:
“The diagnosis of urosepsis is time-consuming and can only be confirmed with blood, during bacteremia, and urine culture results [8]. The process of diagnosing these infections can take from 24 to 72 hours, based on the time required to obtain the culture results. Additionally, there may be false positive results because of contamination, and false negative due to antibiotic administration before admission. Therefore, rapid and efficient diagnostic methods for discriminating urosepsis from complicated UTI are necessary. For a more accurate assessment of the patient, we used procalcitonin [9,10,11] and scores already confirmed by previous studies, such as qSOFA [12], SOFA [13] and CCI [14].”
- The study aim should be clearly explained in the last sentence of the introduction section.
The aim of this study is to evaluate the ability of already existing scores to diagnose, describe the clinical status and predict the evolution of patients with complicated UTI and their risk of progressing to urosepsis.
Materials and Methods:
- "Uroculture" should be urine culture.
Thank you for highlighting this, we have corrected it.
- The study design is not clear and should be revised to have sufficient details.
We have rewritten this including sections as: collected factors, variable definition and patients grouping:
- "blood cultures" how the blood culture could confirm the diagnosis of UTI?
For patients clinically diagnosed with UTI, bacteriological diagnosis can be confirmed by blood culture during bacteremia. So that there are no uncertainties, we have excluded this statement from the main text.
- sample size calculations should be mentioned.
For this purpose, we could not apply a mathematical formula. Instead, we have added to Materials and Methods the next statement: “We conducted a retrospective study including patients diagnosed with complicated urinary tract infections hospitalized in the Urology Department of ”Sfântul Apostol Andrei” County Emergency Clinical Hospital (GCH), a 800-bed general hospital in Galati, Romania, from September 2019 to May 2022. The GCH stands in Galati City which is home for 250,000 citizens. It serves for Galati County region, which has a population of 450,000 people.”
- Study variables:
- In this section, you should mention all collected factors.
We added to this section the following: “ A rigorous clinical examination was performed before admission. Clinical data were collected: heart rate, blood pressure, respiratory rate, PaO2, temperature and Glasgow Coma Scale. Blood and urine samples were collected upon admission, following the International Safety Standards. Complete blood count (CBC), total bilirubin, creatinine and PCT levels were checked upon admission to the hospital. PCT levels were measured using the automatic analyzer VIDAS BRAHMS PCT package insert, according to the manufacturer's instructions. The lower limit of detection of the assay was 0.05 ng/mL and the functional assay sensitivity was 0.09 ng/mL. Demographics, clinical and laboratory findings and diagnostics were recorded. All patients’ medical records were reviewed, and the relevant clinical and biological data were collected. All these data were systematized using qSOFA, SOFA scores and CCI.”
- You may add a new section with the title "variable definition" and you can define any factor involved in your study.
A new section was added “Variable definition” that describe variable used in our study.
- The study outcome should be mentioned.
The aim of this study is to evaluate the ability of already existing scores to diagnose, describe the clinical status and predict the evolution of patients with complicated UTI and their risk of progressing to urosepsis.
Statistical analysis: Describe statistical methods in detail, with emphasis on the statistical strategy used to analyze the data.
- "The variables that have a Gaussian distribution" revise it to quantitive variables, etc…
- The tests used in comparison should be mentioned.
The “Statistical analysis” section was rewritten.
The continuous variables were expressed by descriptive statistics as mean±standard deviation (SD) or median and interquartile range [IQR (Q1-Q3)], while the categorical variables were summarized by absolute and relative frequencies. All continuous variables were checked for normality using the Kolmogorov–Smirnoff test. Descriptive statistics were analyzed through the prism of deceased status, SRIS, Sepsis, Septic Shock. The variables that have a Gaussian distribution (ex. age), were interpreted by means and standard deviation (SD) and the students’ t-test was applied. Variables without a Gaussian distribution: leukocytes, procalcitonin, SOFA, qSOFA, intensive care days, hospitalization days and hospitalization costs were analyzed by median and interquartile range [IQR (Q1-Q3)] and the Mann Whitney test was applied. The correlation between quantitative variables was assessed using Spearmans’ rho correlation, when appropriate. A ROC curve analysis was performed in order to evaluate discriminant accuracy and to find the cut-off values for studied variables. The cutoff level for each variable depending on the analyzed group represents the level for which the best values for sensitivity (the ability to correctly identify the positive diagnosis) and specificity (the ability to correctly identify the negative diagnosis) are simultaneously identified. For all two-sided statistical tests, the significance was achieved if estimated significance level p-value≤ 0.05. Statistical analysis was performed using the MedCalc Soft-ware, Version 12.5.0.0.
Results:
- This section should be started by Giving characteristics of study participants (eg demographic, clinical, social). Please revise this section.
We have added the following: “A total of 174 patients with complicated UTIs were enrolled in this study. The average age was 61.4±15.9 years (mean±SD), 116 (66.7%) patients were from the urban environment and 107 (61.5%) patients were male.”
- All tables' names should be revised.
- Table 1:
- The WBC count is a part of SIRS. What is your aim in mentioning it in table 1?
Given the fact that clinicians tend to give great importance to WBC count when evaluating a septic patient, we wanted to compare their predictive value with the other variables.
We added a paragraph in the discussion section on this issue also: “WBC have been shown to be of low importance in evaluating patients with urosepsis and their mortality risk. Even if WBC can raise an alarm signal regarding the patient's clinical status, being a statistically significant marker for patients in the Septic Shock group, they are not precise in the differential diagnosis between SIRS and Sepsis. And moreover, comparing WBC with the other variables, it proved to be the least accurate in defining the disease. This is an important statement given that clinicians tend to place too much importance in day-to-day practice on the WBC count when evaluating a septic patient.”
- Mean and SD should be in one column.
We have addressed this issue, we merged Table 1 and 2 and redesign it.
- In the (mean 16911.41) and (SD 7792.449), the SC variation is rich to the half value of the mean number, it may due to high variation or error in data analysis. Please review all results.
This happened because leukocytes were wrongly considered to have a normal distribution. We checked for normality using the Kolmogorov–Smirnoff test. Leukocytes proved to have a non-normal distribution; thus, it was described by median and IQR, and Mann Whitney test was applied to analyze it.
- Avoid mentioning "we" in this section.
Thank you for the correction, we rewrote this section and tried to do our best in respecting all highlighted problems.
- Please mention the result of all mentioned factors in your method.
We addressed more attention to economical variables that were neglected in the first draft.
“ The number of intensive care days required for treating these patients was directly proportional to case severity. This was found to be statistically significant for all groups (p<0.001) (Table 1).
Analyzing the number of hospitalization days [median (IQR)], we observed an increase depending on disease severity: 5.0 (3.0-8.0) for the SIRS, 9.0 (6.0-13.7) for the Sepsis and 15.0 (8.0-22.5) for the Septic Shock, this do not apply to the Deceased group [10.0 (6.25-18.7)] p=0.22 (Table 1).
At the same time, increased hospitalization costs [median (IQR)] can be observed in the case of deceased patients [10855.0 (6752.5-24053.2) Lei, p<0.001] and those with septic shock [14704.5 (7357.0-26103.5) Lei, p<0.001] and statistically significantly lower in the case of those with SIRS [2863.0 (1247.0-6833.0) Lei, p<0.001] (Table 1).
Given the fact that the variables that describe the patient from an economic point of view are theoretically interdependent, we used Spearman’s rho to highlight whether this also applies for the patients in our study. Comparing intensive care day and hospitalization days with hospitalization costs it returned p<0.001 in both cases, thus confirming this hypothesis.”
- This table may revise to include and analyze all factors of your study (You may mention all factors in rows and the grouping in columns).
- Table 2: It is hard to follow, please revise it.
We merged Table 1 and 2 and redesign it as suggested.
- The presentation of the obtained result is poorly written and needs to be rewritten with more careful preparation.
We rewrote this section with more careful preparation.
Discussion:
- This section is poorly written. For studies, your Discussion section should first reiterate briefly the results, then move to a discussion of your main findings, and finally, move to wider topics and comparison of your study with other research.
We added to this section and tried to compare our results with other similar research.
- The limitation of your study should be mentioned.
This study was a retrospective study and therefore, bias was likely. Moreover, the scores used were calculated at admission, did not have access to furthermore clinical data necessary to calculate other important scores for our study, such as National Early Warning Score (NEWS), Modified Early Warning Score (MEWS), etc. The data were collected from a single center. As such, it could differ from those of other centers. Additional prospective studies with larger populations involving multiple centers are necessary to validate our conclusion.
Conclusions:
- This section should summarize your main arguments and important findings. Please revised this section.
Although SOFA and qSOFA scores adequately describe patients with septic shock and they are independent prognostic predictors of mortality, they fail to be accurate in diagnosing sepsis. These scores should not replace conventional triage protocol. In our study PCT has proven to be a disease-defining marker and an independent prognostic predictor of mortality. Patients with important comorbidities, CCI greater than 10, should be treated more aggressively because of increased mortality.
References
- Please correct your references according to the authors' guidelines.
- Journal names should be abbreviated to the Journals Database section in PubMed (http://www.ncbi.nlm.nih.gov/nlmcatalog/journals).
We have corrected the errors found in this section.
Reviewer 2 Report
The article have a nice and interesting point of view, yet there are several aspects that need to be improved.
English language and style need to be extensively improved and some sentence need more paraphrasing as it is still traceable for plagiarism in some of the sentences.
Comment 1:
The title is need to be more concise and focused, it can be
Comment 2:
Please provide the aim of the study in introduction
Comment 3:
Line 71: "uroculture" seems like not a standard English terms
Comment 4:
Line 102: please state clearly that it is Glasgow Coma Scale not just Glasgow
Comment 5:
Table 1 and 2 are a little bit difficult to interpret, I suggest to simplify or show it as diagram and put the tables in the suplementary data
The title of those table also unclear
The standard name for the statistic test in table 1 is "student's t test"
In table 2, you usually only need to put the median (the maximum and minimum number is not needed, a box plot also good to represent this kind of data
Comment 6:
Line 73-74 please elaborate the "etc", as it is the inclusion criteria, you need to be precise.
Comment 7:
Please choose between the ROC curve or the table for picture 1-4/table 3-6, those table can be added to suplementary data
Comment 8:
Lines 180-188 please provide reference.
Comment 9:
Line 192, please provide the spearman correlation result in the result section and elaborate with previous studies.
Comment 10:
Line 218, please elaborate more about Procalcitonin from previous studies
Comment 11:
Conclusion need to address the aim of the study
Author Response
Comments and Suggestions for Authors
The article have a nice and interesting point of view, yet there are several aspects that need to be improved.
English language and style need to be extensively improved and some sentence need more paraphrasing as it is still traceable for plagiarism in some of the sentences.
Comment 1:
The title is need to be more concise and focused, it can be
We have modified it into: “The role of biomarkers and scores in describing urosepsis”
Comment 2:
Please provide the aim of the study in introduction.
The aim of this study is to evaluate the ability of already existing scores to diagnose, describe the clinical status and predict the evolution of patients with complicated UTI and their risk of progressing to urosepsis.
Comment 3:
Line 71: "uroculture" seems like not a standard English terms.
Thank you for highlighting this, we have corrected it and used the proper term urine culture.
Comment 4:
Line 102: please state clearly that it is Glasgow Coma Scale not just Glasgow
We have used the proper terminology.
Comment 5:
Table 1 and 2 are a little bit difficult to interpret, I suggest to simplify or show it as diagram and put the tables in the suplementary data
We have addressed this issue, we merged Table 1 and 2 and redesign by mentioning all factors in rows and the grouping in columns.
The title of those table also unclear
We renamed it: “Bivariate analysis of variables according to studied groups”
The standard name for the statistic test in table 1 is "student's t test"
Thank you for highlighting this, we have corrected it.
In table 2, you usually only need to put the median (the maximum and minimum number is not needed, a box plot also good to represent this kind of data
This was also reviewed, we described variables with non-normal distribution by median and IQR.
Comment 6:
Line 73-74 please elaborate the "etc", as it is the inclusion criteria, you need to be precise.
Thank you for highlighting this, we have corrected it.
Comment 7:
Please choose between the ROC curve or the table for picture 1-4/table 3-6, those table can be added to suplementary data
We chose to keep the ROC curve figures due to their ability to be more easily understood.
However, we believe that the tables also contain important information, difficult to reproduce in the text, which attests to the rigors of the research. Trying to find a solution to keep them in the main text, we merged all the tables into one. If this option is not acceptable, we will transfer it to the supplementary data section.
Comment 8:
Lines 180-188 please provide reference.
Thank you for highlighting this, we have corrected it.
Comment 9:
Line 192, please provide the spearman correlation result in the result section and elaborate with previous studies.
We addressed to this issue by mentioning about Spearman’s rho in the statistical section and in the results section.
Comment 10:
Line 218, please elaborate more about Procalcitonin from previous studies
We added the following statement:” PCT accurately predicts the presence of bacteremia and bacterial load in patients with complicated UTI [26]. Some studies have shown that a PCT >2 ng/mL has >90% specificity for sepsis or the progression to sepsis [27]. In our study the cutoff value for PCT in discriminating sepsis was 4.8. The PCT levels were higher in our study for the following possible reasons: most of the patients with UTI were infected with gram-negative organisms, which causes higher peak PCT values than infections caused by gram-positive organisms [28] different observation times may have resulted in different PCT optimal cutoff values to diagnose sepsis [29] different test kits and methods may have been used [30] “
Comment 11:
Conclusion need to address the aim of the study
Although SOFA and qSOFA scores adequately describe patients with septic shock and they are independent prognostic predictors of mortality, they fail to be accurate in diagnosing sepsis. These scores should not replace conventional triage protocol. In our study PCT has proven to be a disease-defining marker and an independent prognostic predictor of mortality. Patients with important comorbidities, CCI greater than 10, should be treated more aggressively because of increased mortality.

Reviewer 3 Report
Thank you for the opportunity to review this work. This manuscript is a prospective observational study to demonstrate the ability of the predictive scoring systems. Detailed comments about this study are as follows:
-Nowadays, there are various scoring systems to evaluate septic patients, such as qSOFA, National Early Warning Score (NEWS), and REDS score. Moreover, several recent studies revealed that NEWS is more accurate for detecting sepsis or septic shock than SIRS or qSOFA score. Why did the author not include the NEWS in this comparison?
-Why did the authors use the SIRS criteria as part of this comparison? In 2016, the European Society of Intensive Care Medicine and the Society of Critical Care Medicine (SCCM) proposed Sepsis-3 (The Third International Consensus Definitions for Sepsis and Septic Shock), which is a new definition for sepsis. This new definition excluded SIRS criteria to consider as sepsis because SIRS criteria is nonspecific as any life-threatening organ dysfunction caused by the dysregulated host response to infection. Meanwhile, the new "sepsis" definition is a “life-threatening organ dysfunction caused by a dysregulated host response to infection.”
-The authors mentioned the related keywords of this manuscript as “urosepsis; SOFA; qSOFA scores; CCI.” However, those could not be found in the Medical Subject Headings (MeSH) (available from https://meshb.nlm.nih.gov). Providing the other keywords might be more suitable.
-In the statistical analyses section, the non-normally distributed data should be present as a median and interquartile range rather than the minimum and maximum in order to eliminate the outlier.
-The author indicated that “The variables that have a Gaussian distribution: age and leukocytes, were interpreted by means and standard deviation (SD) and the Student test was applied.” A term “student’s t-test” is more suitable than “Student test.”
-How to determine the Gaussian distribution? Please provide the method used, such as visualization or normality test.
-How about the sample size estimation in this study? If available, it should be mentioned.
-In the statistical analyses section, how did to demonstrate the difference in AUC among the following variable (procalcitonin, leukocytes, CCI, SOFA, qSOFA)? AUC for each variable is found in the result section and Tables 3, 4, and 5 without statistical evidence of their difference. Also, the author mentioned only sentences as follows without statistical test of those AUCs: “Comparing the ROC curves for the variables of interest, a very good specificity and sensitivity can be identified in predicting the risk of death for procalcitonin (69.57%, 77.33%), the SOFA (91.33%, 76.82%), qSOFA (91.30%, 74.17%) scores and CCI (65.22%, 88.74%). The AUC value being best for qSOFA 90.3%” Please provide the statistical evidence of the difference among those AUCs (in comparing the AUCs).
-Figure 1 (Picture 1 Pairwise comparison of ROC curves for the Deceased group 4) in the manuscript shows a residual at the left border of its figure. Please erase it from that figure. Moreover, it should provide the vertical axis name as “Sensitivity.”
-Figure 1 and 2 (Picture 1 and 2) should be stated the full term of the abbreviation of CCI, SOFA, and qSOFA.
-In Table 1, there was an unusual term for “Test Student.” Another word or phrase should be used instead.
-In Table 1, the column design should be improved for easier comparison. For example, “Survival,” “Deceased,” and “p-value (comparison of means)” might be used in the column; meanwhile, “Age” and “Leukocyte” leave it in the row divided by the subheading of the row as overall, SIRS, Sepsis, Septic shock. The value of the mean, SD, should be mentioned in each cell.
-Also, Table 2 design should be improved for easier comparison.
-The decimal of the p-value should be consistent throughout the manuscript. For example, the authors present the p-value in Table 1 as “0.002, …, 0.0001, …” I suggest changing it into “0.002, …, <0.001, …” or “<0.01, …, <0.01, …” Also, please change it in the same manner throughout the manuscript.
-Please correct the word “miniumum” in line 111.
-Please correct the word “SRIS” in Table 2 and line 113.
-Please provide the 95% CI of the sensitivity, and specificity in Tables 3, 4, and 5.
-Please correct the subscript of many places, such as “PaCO2” (page 2, line 79)
-Please correct the superscript of many places, such as “/mm3” (page 2, line 80), “x106/L” (page 4, lines 133-134).
-I suggest moving the sentence (lines 144-149): “The predictability of discriminating urosepsis stages was assessed by using the area under the ROC curve (AUC). For this purpose, the cutoff level was calculated for each variable. The cutoff level for each variable depending on the analyzed group represents the level for which the best values for sensitivity (the ability to correctly identify the positive diagnosis) and specificity (the ability to correctly identify the negative diagnosis) are simultaneously identified.” from the result section to the method section.
-It might remove the phrase “Table 3 Pair-wise comparison of ROC curves for the Deceased group” in lines 150-151.
-In the discussion section, the authors presented as follows: "Using Spearman's rank correlation coefficient, we demonstrate a statistically significant correlation between age and CCI (p<0.001)." “Using Spearman's rank correlation coefficient, a statistically significant correlation between SOFA and the hospitalization days and costs (p<0.001) was demonstrated.” However, those finding was not mentioned in the result sections. Therefore, those findings should state in the result instead and then discussed in the discussion section. Furthermore, if Spearman's rank correlation was used in the statistical analysis, it should be stated in the method section.
-The authors indicated, “All the variables have an increased specificity, but a low sensitivity, this meaning a risk of false positive diagnosis.” However, it might be a fault because sensitivity is calculated from “true positive/(true positive + false negative).” So, low sensitivity is found in low true positive and high false negative conditions. Therefore, a test with low sensitivity can result in a false negative rather than a false positive.
-What is the limitation of this study?
Author Response
Comments and Suggestions for Authors
Thank you for the opportunity to review this work. This manuscript is a prospective observational study to demonstrate the ability of the predictive scoring systems. Detailed comments about this study are as follows:
Thank you for taking the time to review our work. We have taken into account all your suggestions and hope that we have properly solved all the issues.
-Nowadays, there are various scoring systems to evaluate septic patients, such as qSOFA, National Early Warning Score (NEWS), and REDS score. Moreover, several recent studies revealed that NEWS is more accurate for detecting sepsis or septic shock than SIRS or qSOFA score. Why did the author not include the NEWS in this comparison?
Given the fact that our study is a retrospective one and that the scores used in our work were calculated at admission, we did not have the necessary clinical data to be able to calculate new scores without risking bias. We are aware of the mentioned scores; we are trying to introduce them in daily practice and in future research.
-Why did the authors use the SIRS criteria as part of this comparison? In 2016, the European Society of Intensive Care Medicine and the Society of Critical Care Medicine (SCCM) proposed Sepsis-3 (The Third International Consensus Definitions for Sepsis and Septic Shock), which is a new definition for sepsis. This new definition excluded SIRS criteria to consider as sepsis because SIRS criteria is nonspecific as any life-threatening organ dysfunction caused by the dysregulated host response to infection. Meanwhile, the new "sepsis" definition is a “life-threatening organ dysfunction caused by a dysregulated host response to infection.”
The methods used in our study were not clearly stated. We have introduced a new subchapter " Patient grouping according to the disease stage " that defines the grouping. Patients in the SIRS group were patients with complicated UTIs who meet SIRS criteria but did not have sepsis on admission according to the new sepsis definition.
-The authors mentioned the related keywords of this manuscript as “urosepsis; SOFA; qSOFA scores; CCI.” However, those could not be found in the Medical Subject Headings (MeSH) (available from https://meshb.nlm.nih.gov). Providing the other keywords might be more suitable.
Using MeSH database, we have chosen the following keywords: urinary tract infection, sepsis, diagnosis, biomarkers, sensitivity and specificity.
-In the statistical analyses section, the non-normally distributed data should be present as a median and interquartile range rather than the minimum and maximum in order to eliminate the outlier.
We have addressed to the problem, as stated in the Results section, Table 1; all non-normal distributed data was described by median and IQR.
-The author indicated that “The variables that have a Gaussian distribution: age and leukocytes, were interpreted by means and standard deviation (SD) and the Student test was applied.” A term “student’s t-test” is more suitable than “Student test.”
Thank you for highlighting this, we have modified it.
-How to determine the Gaussian distribution? Please provide the method used, such as visualization or normality test.
All continuous variables were checked for normality using the Kolmogorov–Smirnoff test. We have stated this in the main text also.
-How about the sample size estimation in this study? If available, it should be mentioned.
For this purpose, we could not apply a mathematical formula. Instead, we have added to Materials and Methods the next statement: “We conducted a retrospective study including patients diagnosed with complicated urinary tract infections hospitalized in the Urology Department of ”Sfântul Apostol Andrei” County Emergency Clinical Hospital (GCH), a 800-bed general hospital in Galati, Romania, from September 2019 to May 2022. The GCH stands in Galati City which is home for 250,000 citizens. It serves for Galati County region, which has a population of 450,000 people.”
-In the statistical analyses section, how did to demonstrate the difference in AUC among the following variable (procalcitonin, leukocytes, CCI, SOFA, qSOFA)? AUC for each variable is found in the result section and Tables 3, 4, and 5 without statistical evidence of their difference.
The AUC was calculated individually for each variable, but the difference between the AUCs was also calculated separately. In the ROC model, the 5 markers were entered and implicitly the difference between the AUC was calculated.
Also, the author mentioned only sentences as follows without statistical test of those AUCs: “Comparing the ROC curves for the variables of interest, a very good specificity and sensitivity can be identified in predicting the risk of death for procalcitonin (69.57%, 77.33%), the SOFA (91.33%, 76.82%), qSOFA (91.30%, 74.17%) scores and CCI (65.22%, 88.74%). The AUC value being best for qSOFA 90.3%” Please provide the statistical evidence of the difference among those AUCs (in comparing the AUCs).
We compare all the variables with each other, one by one, for each individual group. These data are systematized in laborious tables that cannot be incorporated into the text. With your consent we will upload these data in the addendum.
E.g.: Pairwise comparison of ROC curves for Deceased group
|
procalcitonina ~ leucocite |
|
|
Difference between areas |
0.196 |
|
Standard Error c |
0.0916 |
|
95% Confidence Interval |
0.0162 to 0.375 |
|
z statistic |
2.137 |
|
Significance level |
P = 0.033 |
|
procalcitonina ~ CCI |
|
|
Difference between areas |
0.0907 |
|
Standard Error c |
0.0703 |
|
95% Confidence Interval |
-0.0472 to 0.229 |
|
z statistic |
1.290 |
|
Significance level |
P = 0.197 |
|
procalcitonina ~ SOFA |
|
|
Difference between areas |
0.121 |
|
Standard Error c |
0.0536 |
|
95% Confidence Interval |
0.0160 to 0.226 |
|
z statistic |
2.258 |
|
Significance level |
P = 0.024 |
|
procalcitonina ~ qSOFA |
|
|
Difference between areas |
0.131 |
|
Standard Error c |
0.0595 |
|
95% Confidence Interval |
0.0142 to 0.247 |
|
z statistic |
2.199 |
|
Significance level |
P = 0.028 |
|
leucocite ~ CCI |
|
|
Difference between areas |
0.287 |
|
Standard Error c |
0.0872 |
|
95% Confidence Interval |
0.116 to 0.457 |
|
z statistic |
3.287 |
|
Significance level |
P = 0.001 |
|
leucocite ~ SOFA |
|
|
Difference between areas |
0.317 |
|
Standard Error c |
0.0648 |
|
95% Confidence Interval |
0.190 to 0.444 |
|
z statistic |
4.888 |
|
Significance level |
P < 0.001 |
|
leucocite ~ qSOFA |
|
|
Difference between areas |
0.327 |
|
Standard Error c |
0.0635 |
|
95% Confidence Interval |
0.202 to 0.451 |
|
z statistic |
5.142 |
|
Significance level |
P < 0.001 |
|
CCI ~ SOFA |
|
|
Difference between areas |
0.0304 |
|
Standard Error c |
0.0482 |
|
95% Confidence Interval |
-0.0640 to 0.125 |
|
z statistic |
0.631 |
|
Significance level |
P = 0.528 |
|
CCI ~ qSOFA |
|
|
Difference between areas |
0.0401 |
|
Standard Error c |
0.0533 |
|
95% Confidence Interval |
-0.0643 to 0.145 |
|
z statistic |
0.753 |
|
Significance level |
P = 0.451 |
|
SOFA ~ qSOFA |
|
|
Difference between areas |
0.00971 |
|
Standard Error c |
0.0268 |
|
95% Confidence Interval |
-0.0427 to 0.0622 |
|
z statistic |
0.363 |
|
Significance level |
P = 0.717 |
-Figure 1 (Picture 1 Pairwise comparison of ROC curves for the Deceased group 4) in the manuscript shows a residual at the left border of its figure. Please erase it from that figure. Moreover, it should provide the vertical axis name as “Sensitivity.”
We solved this framing problem.
-Figure 1 and 2 (Picture 1 and 2) should be stated the full term of the abbreviation of CCI, SOFA, and qSOFA.
We used a footnote for these abbreviations
-In Table 1, there was an unusual term for “Test Student.” Another word or phrase should be used instead.
Was replaced by the term "student's t-test"
-In Table 1, the column design should be improved for easier comparison. For example, “Survival,” “Deceased,” and “p-value (comparison of means)” might be used in the column; meanwhile, “Age” and “Leukocyte” leave it in the row divided by the subheading of the row as overall, SIRS, Sepsis, Septic shock. The value of the mean, SD, should be mentioned in each cell.
-Also, Table 2 design should be improved for easier comparison.
We have addressed this issue, we merged Table 1 and 2 and redesign it.
-The decimal of the p-value should be consistent throughout the manuscript. For example, the authors present the p-value in Table 1 as “0.002, …, 0.0001, …” I suggest changing it into “0.002, …, <0.001, …” or “<0.01, …, <0.01, …” Also, please change it in the same manner throughout the manuscript.
All values of p were modified according to your suggestion.
-Please correct the word “miniumum” in line 111.
Thank you for highlighting this, we have corrected it.
-Please correct the word “SRIS” in Table 2 and line 113.
Thank you for highlighting this, we have corrected it.
-Please provide the 95% CI of the sensitivity, and specificity in Tables 3, 4, and 5.
We merged tables 3-6 at the reviewer's recommendation and added 95% CI.
-Please correct the subscript of many places, such as “PaCO2” (page 2, line 79)
Thank you for highlighting this, we have corrected it.
-Please correct the superscript of many places, such as “/mm3” (page 2, line 80), “x106/L” (page 4, lines 133-134).
Thank you for highlighting this, we have corrected it.
-I suggest moving the sentence (lines 144-149): “The predictability of discriminating urosepsis stages was assessed by using the area under the ROC curve (AUC). For this purpose, the cutoff level was calculated for each variable. The cutoff level for each variable depending on the analyzed group represents the level for which the best values for sensitivity (the ability to correctly identify the positive diagnosis) and specificity (the ability to correctly identify the negative diagnosis) are simultaneously identified.” from the result section to the method section.
We moved this sentence to the method section.
-It might remove the phrase “Table 3 Pair-wise comparison of ROC curves for the Deceased group” in lines 150-151.
We solved this problem.
-In the discussion section, the authors presented as follows: "Using Spearman's rank correlation coefficient, we demonstrate a statistically significant correlation between age and CCI (p<0.001)." “Using Spearman's rank correlation coefficient, a statistically significant correlation between SOFA and the hospitalization days and costs (p<0.001) was demonstrated.” However, those finding was not mentioned in the result sections. Therefore, those findings should state in the result instead and then discussed in the discussion section. Furthermore, if Spearman's rank correlation was used in the statistical analysis, it should be stated in the method section.
We mentioned Spearman's rho in the methods section and in the results.
-The authors indicated, “All the variables have an increased specificity, but a low sensitivity, this meaning a risk of false positive diagnosis.” However, it might be a fault because sensitivity is calculated from “true positive/(true positive + false negative).” So, low sensitivity is found in low true positive and high false negative conditions. Therefore, a test with low sensitivity can result in a false negative rather than a false positive.
Thank you for highlighting this.
We have corrected this issue:
“All the variables have an increased specificity, but a low sensitivity, this translates into a risk of false negative diagnosis leading to the lack of adequate treatment”
-What is the limitation of this study?
This study also had several limitations. It was a retrospective study and therefore, bias was likely. Moreover, the scores used were calculated at admission, did not have access to furthermore clinical data necessary to calculate other important scores for our study, such as National Early Warning Score (NEWS), Modified Early Warning Score (MEWS), etc. The data were collected from a single center. As such, it could differ from those of other centers. Additional prospective studies with larger populations involving multiple centers are necessary to validate our conclusion.

Round 2
Reviewer 1 Report
Dear author
This is an interesting study methodology. However, there are hardly any paragraphs (even in the abstract). The manuscript still needs scientific English editing and should be rewritten by scientific medical specialized experts. The abstract, method, result, and tables should be prepared according to the journal guidelines, avoiding any long unimportant sentences with a proper scientific presentation. The presentation data in the tables also needs careful revision.
Author Response
Thank you for taking the time to review our work. Your suggestions have been taken into consideration and we have tried our best to resolve all the issues. We have tried to paraphrase correctly, adhere to proper drafting rules, and have accessed the services of a medical scientific expert. We also revised the References section using Zotero Software
Reviewer 2 Report
The article has already improved so much.
Author Response
Thank you for taking the time to review our article. Thank you for the rigor you showed in the previously presented suggestions that helped us significantly improve our work. We hope that in the future our collaboration to have the same results.
Reviewer 3 Report
-The authors have modified the text following the reviewers' suggestions. I have some minor suggestions.
-Figure 1 (also Figures 2, 3, 4) still shows a residual at the left border of their figures in my reviewing manuscript. Please erase it from those figures. Moreover, it should provide the vertical axis name as “Sensitivity” close to the vertical axis line.
-Please add a semicolon between “qSOFA=quick SOFA” and “CCI= Charlson Comorbidity Index” to separate them in the Table footnote in Figures 1, 2, 3, and 4.
-There is a redundancy of the abbreviation “SD” in lines 160 and 165. Please remove the last one.
